# Effects of Carbohydrate and Protein Administration by Food Items on Strength Response after Training in Stable COPD

**DOI:** 10.3390/nu14173565

**Published:** 2022-08-30

**Authors:** Andrea Huhn, Ulrich Flenker, Patrick Diel

**Affiliations:** 1Zwanzig-Neun-Fünf Essen GmbH, 45127 Essen, Germany; 2Department of Molecular and Cellular SportsMedicine, Institute for Cardiovascular Research and Sports Medicine, German Sports University, 50333 Cologne, Germany

**Keywords:** COPD, strength training, protein, regeneration, chronic lung disease, maximum strength

## Abstract

Background: Chronic obstructive pulmonary disease (COPD) is one of the world’s most common diseases and reasons for death. Systemic consequences, especially reduced muscle strength, muscle mass and muscle function, are common and contribute to worsening prognosis and increasing morbidity and mortality. There is strong evidence that physical activity and strength training are effective in prolonging life and lead to better quality of life. Numerous studies have shown that ingestion of protein and carbohydrates after strength training can increase regeneration of strength in young athletes. Recently, we demonstrated that the same effect can be achieved with these macronutrients administered in a meal. Until now, it is not clear if patients with COPD, integrated in regular physical training, respond similarly. Methods: Prescribed strength training, consisting of two sets circular training with machines for big muscle groups was supplemented with a meal rich in protein and carbohydrates. Changes in maximum strength after 24 h were investigated to find out about the impact of this meal on physical capacity. A pilot study was conducted with pragmatic cross-over design. Results: With nutritive intervention, strength in both knee extensor and chest press were significantly higher than in control training. Conclusion: The study showed beneficial effects for the intake of protein and carbohydrates in changes in maximum strength. For now, the underlying mechanism remains unclear. Clinical relevance needs further research. The study design and study protocol can be used for further studies with only small adaptions.

## 1. Introduction

At 5.8%, chronic obstructive pulmonary disease (COPD) accounts for one of the most common reasons for death in the world [1]. It is expected to increase in prevalence during the next decades [2]. Severity of the disease can be described by the GOLD stages (GOLD I–IV), which are defined by airflow limitation and measured with FEV_1_. One crucial consequence of COPD is limb muscle dysfunction which fundamentally determines prognosis in terms of exercise tolerance, quality of life and mortality [3]. Muscle dysfunction occurs in all stages of COPD but increases with higher stages [3]. One-third of patients with COPD are considered to have functional impairment [4]. Treatment of muscle dysfunction by increasing recovery of strength is therefore the key motivation of the conducted pilot study.

It is still discussed whether muscle dysfunction in COPD is the result of physical inactivity or if disease itself affects muscles [3]. Underlying mechanisms, including systemic inflammation, oxidative stress, and hypoxemia, might be involved [3]. Insulin resistance [5], menopause [6], or use of glucocorticoids [7] might change hormonal balance and contribute in muscle dysfunction. The etiology of muscle dysfunction has clinical impact on treatment. If only deconditioning is the reason for muscle dysfunction, this process can be reversible with adequate training. If muscle dysfunction is from a myopathic cause, treatment of these causes must be included [3]. It is therefore not clear if patients with COPD react similar to training interventions as healthy people. It might also be possible that the training effect is diminished due to low training tolerance and therefore lower impact [3].

Nevertheless, the general effect of training to enhance people’s physical capacity and reduce morbidity and mortality has been validated several times and recommendations can be found in leading guidelines for COPD [7,8,9,10]. Training can be optimized using high-intensity stimulus [11] and the combination of strength and endurance training [12]. Strength training is very suitable since this type of exercise does not provoke dyspnea, which is usually a key limitation of activity [13].

Nutritional interventions might enhance physical restoration and potentiate exercise response, leading to more benefit for the patient’s physical capacity [14]. The combination of protein and carbohydrates after strength training is a common tool to expedite muscle recovery and increase anabolic response [14,15]. Protein should contain high essential amino acids (EAA) and should be consumed in combination with carbohydrates during the first 30 min after exercise [15]. Apart from changes in blood parameter that indicate muscle damage and inflammation, maximum strength is less affected after training with this kind of nutritional intervention than without [16].

This knowledge is mostly based on studies with young healthy athletes. Until now, evidence for nutritional intervention in COPD is very controversial due to high variance in training, nutritional interventions, and testing modalities as well as high intra-individual differences in patients. Instead of training with specific nutritional interventions, the literature focusses on weight gain or general increase in protein intake [17,18,19]. Most changes with interventions like this are an increase of fat mass and not muscle mass [20]. Mostly, untrained individuals are investigated. Since training is most effective intervention in muscle dysfunction, it could cover the add-on effects [21]. Functional parameters are not often investigated, even though these are suggested to be the main patient outcome. Few studies measure changes in strength and have contrasting results [19]. According to recent knowledge, EEA are the most effective nutritional interventions, compared to other nutrients, to increase muscle function in COPD [22]. Evidence to optimize training stimulus is still needed considering individual health and physical conditions of patients that might influence response [13].

Our research team has recently demonstrated a physiological and functional response and benefit for consumption of protein and carbohydrates after high-intensity endurance exercises in healthy young individuals. In this study, we could also demonstrate that uptake of carbohydrates and protein by food could be as effective in reducing skeletal muscle damage and improving skeletal muscle regeneration as by drinking whey protein/glucose shakes [16]. The effect of increased strength gain with protein and carbohydrates after strength training is also evident [3]. A transfer to patients with COPD is not sure as the disease goes ahead with many physiological changes that can lead to muscle dysfunction and impaired training response [23,24]. Nevertheless, a general training response is still evident [3]. Additionally, the response to protein and carbohydrates is suggested to be impaired in COPD [24,25]. Insulin resistance, reduced sensitivity to amino acids, and a lack of plasma branched-chain amino acids [25] might affect response [5,26]. For this reason, timing and amount of protein might become more important in the elderly [14].

To overcome the lack of evidence for patients with COPD, this pilot study focuses on short-term effects of nutritional intervention to strength response. The results might give more information about the strength response of resistance training and pro-regenerative support. This should optimize the treatment of limb muscle dysfunction. It was hypothesized that maximum strength and functional capacity decrease 24 h after strength training and that decrease is diminished with nutritional intervention. Moreover, it was suggested, that the results differ between subgroups of COPD in terms of training intensity and nutritional state.

Patients with stable COPD were tested in functional response of a combinatory exercise and nutritive intervention in cross-over design. Using high-intensity resistance training and a meal rich in protein and carbohydrates, changes in maximum strength in knee extensor and chest press as well as functional capacity were measured.

## 2. Materials and Methods

Study was ethically approved by German Sports University Cologne. All participants provided written informed consent prior to their participation. The study included patients with COPD (GOLD I–IV) with medical prescription of strength training with devices (“gerätegestützte Krankengymnastik”, covered by medical insurance in Germany) and minimum experience time of 4 weeks. Subjects were personally addressed during their usual training. Patients with contraindications for strength training (acute infection, decompensated heart insufficiency, osteoporosis with risk of fracture) or intolerance of applied food were excluded. Further exclusion criteria were the use of any functional food, supplements, or protein shakes for improvement in exercise capacity, insulin-dependent or instable diabetes, and neurologic comorbidities. Subjects maintained their usual diet and exercise habits during the whole study period.

Anthropometric data can be found in Appendix A. Mean age was 64.7 ± 7.3. Most patients had COPD GOLD II, followed by each two cases of stage II and IV and one case of GOLD I. Mean FEV_1_ was 53.7 ± 19.7. Categorization after ABCD scheme was not possible due to very different results in COPD Assessment Test (CAT) and modified Medical Research Council (mMRC), which both determine scores. Total multidimensional disease classification with BODE Index varied between 4 and 12 with a median of 5 and a mean of 6.4. In 6-Minute Walk Test (6MWT), mean was 473.4 ± 77.8 m, and one subject had a capacity which is referred to high risk of mortality (≤350 m) [27]. Mean BMI was 27.1 ± 5.9. Two participants had a low BMI (BMI ≤ 21), and one was underweight (≤18.5).

Study was planned and conducted in cross-over design with alternated quasi-randomization (See Figure 1).

Both interventions and post-tests took place at same time of day (±60 min) after individual preference. Ten patients with COPD performed an intense supervised strength training according to their usual protocol in physiotherapeutic setting. Testing was performed before and 24 h after each training. Patients started with a ten-minute general warm-up on individual level (70% of usual training intensity) on ergometer or treadmill. After a five-minute break, customization with knee extensor was conducted using ten repetitions with training intensity followed by one-minute break and maximum strength test. After one minute of rest, the same procedure was performed with chest press. Another five-minute rest was given until functional capacity was assessed, followed again by five minutes’ break. The same investigation was conducted 24 h after pre-test, including general and specific warm-up and testing.

Training intensity and range of motion was not changed from usual training to guarantee familiarization of previous trainings. Adjustments were based on training introduction from physiotherapists in the setting and were regularly actualized with the aim of local fatigue after one minute of exercise. All relevant training variables for a non-progressive strength training program can be found in Table 1 [28]. Individual adaptions were possible if program could not be conducted due to physical limitations. Two participants only did one set of training machines, and one of them with a volume of only 40 s due to breathlessness.

Participants had at least ten minutes rest after strength training. Meal was provided with plain water only after one training during the first 30 min after strength training. Nutrients are presented in Table 2.

Patients were asked not to eat or drink anything (except for provided food) but plain water one hour before and two hours after training, independently from nutritional intervention or not. They were asked not to change their usual diet during period of study intervention. A food diary was completed twelve hours before until 24 h after each training. Subjects were asked not to perform further strength training with devices between training interventions. Strenuous activities 12 h before and 24 h after training should be avoided and were recorded. Physical Activity Scores for the Elderly (PASE) was completed for the previous week. A minimum period of 5 days was planned between two trainings. During that time, complete regeneration was supposed. Strength of upper and lower extremities was measured with indirect method of one-repetition maximum (1RM), which is a safe [29] and validated test for patients with COPD [30,31]. After familiarization, concentric weight was doubled to reach a maximum of 1–12 repetitions in full range of motion. If patients had osteoporosis, only 150% of training weight was used. Patients were supervised to exhale during concentric movement to prevent Valsalva maneuver. Verbal encouragement was used.

For knee extensor, starting position was sitting with 90° knee flexion, ending position with 170° extension. For chest press, starting position was calculated 90° shoulder abduction and 20° extension. For ending position, 90° shoulder flexion with submaximal elbow extension was used.

For calculating 1RM, formula of Kemmler was used [32]. Apart from maximum strength as main outcome, functional muscle capacity was measured with one-minute sit-to-stand test (STST). Before warm-up and during the last two minutes of general warm-up, modified Borg CR10 scale for perceived exertion of legs and breath and pulse oximetry was assessed.

Data were analyzed by fitting linear mixed effects models (LME). We were precisely interested in the potential interaction of the training bouts and subsequent alimentation. Therefore, the factors training (levels pre, post) and food (levels treat, control) and their interaction were stipulated a priori as fixed effects in all fitted models. All subjects were investigated repeatedly while they were following their habitual training schedules. Therefore, the additional factor time was introduced to account for individual trends. Time was expressed as days difference from individual enrollment. Random effects were hence tried to model as time grouped by subject. Starting from intercepts only (order 0), the individual effects to time were modeled “bottom-up” by polynomials in increasing order. If the presence of individual trends appeared unlikely, random effects were assumed to be associated with the respective testing session. The fitted models were compared by likelihood statistics. Fundamentally, the model with the lowest Akaike Information Criterion (AIC) was preferred, but only if the likelihood ratio test yielded a *p*-value < 0.05 as compared to the next simpler model. The preferred model then served to evaluate the parameter estimates of the fixed effects (t-statistics) and to calculate the effect size (Cohen’s d). Cohen’s d was approximated as 2·t/DF (1/2). The fulfilment of the model assumptions was tested ex post by residual plots (homoskedasticity) and by Q-Q plots (Gaussianity of the residuals). The software was R in its latest version [33] where the nlme-library [34] was employed to fit the LMEs. For correlation analysis, Spearman rank correlation was used. Responsiveness and food diary was analyzed qualitatively.

## 3. Results

The data of nine subjects was collected for knee extensor and eight for chest press. One dataset was excluded since the patient with psychic comorbidity had a high stress level and could not fulfill the testing modalities. One patient was not assessed in the chest press due to shoulder problems. For all participants, indirect methods between 1–12 was reached with first attempt and showed good reliability at T_1_ and T_3_. Two participants had less than 120% relative quadriceps strength (absolute strength/BMI), which is associated with higher risk of mortality [35]. The mean (SD) was 136.67 (41.24)%, and the median (min; max) was 150 (57; 192)%. The mean concentric training intensity was 44.66% of 1RM for knee extensor and 50.29% for chest press. Absolute and relative quadriceps strength did not correlate with lung function (FEV_1_), multidimensional disease scale (BODE) or mean activity level (PASE).

Strength reduction occurred in zero cases in the leg extensor and chest press after nutritional intervention, while five cases occurred without nutritional intervention.

### 3.1. Quadriceps Strength

Mean change in knee extensor was −0.67 ± 1.14 kg for control and 1.13 ± 1.85 kg for intervention (*n* = 9). The difference between control training and nutritional intervention was significant. During the course of the study, the between-subjects variation of the knee extension was best described by assuming individual linear trends. This model showed the lowest AIC. The fit was significantly improved as compared to the assumption of timely invariant intercepts (*p* = 0.0015, likelihood ratio test). Higher order polynomials yielded insignificant improvements of the fits but clearly increasing for the AICs.

The residual standard deviation of the finally accepted model was ±1.02 kg. The estimates and statistics of the fixed effects are presented in Table 3.

The 95%-confidence interval for the interaction effect is 0.40–3.21 kg.

Figure 2 shows the boxplots of the raw data of the individual fits and of the population fits.

### 3.2. Chest Press

The mean change in the chest press was −0.12 ± 0.90 kg in the control and 1.61 ± 2.18 kg for the intervention (*n* = 8). The difference between the control and intervention was significant. In contrast to the knee extension, the chest press data did not exhibit any identifiable individual trends with sufficient likelihood. Instead, the assumption of individual random effects for the respective training sessions resulted in the lowest AIC. The corresponding model fitted significantly better (*p* = 0.0027) than the simplest model (cf. knee extension).

The residual standard deviation of the finally accepted model was ±0.38 kg. The estimates and statistics of the fixed effects are presented in Table 4.

The 95%-confidence interval for the interaction effect is 0.03–3.44 kg.

Figure 3 shows the boxplots of the raw data of the individual fits and of the population fits.

### 3.3. Other Measurements

For STST, results varied between 7 and 30 with a mean of 19 repetitions (*n* = 8) and were consistent for each participant in terms of pre-term values. STST correlated strongly and significantly (*p* = 0.045; r = 0.72) with absolute strength and non-significantly (*p* = 0.1) with relative strength. Most participants increased the number of repetitions after training.

The food diary only gave an insight into participants’ eating habits since patients were asked to report their food intake in as much detail as possible but without weighing their meals. With this low informative value, it was only analyzed if patients had three full meals during a day, including one warm meal. This was defined as regular and balanced nutrition and was seen in 6 of 9 participants. In the explorative analyses it was striking that, most likely, the remaining subjects had very few and small meals with low protein intake. None of them had low BMI. With and without intervention, three participants were fasting before training. Smoking status and number of cigarettes consumed was assessed even though short-term changes in maximum strength are not expected in strength training [6]. One participant was smoker with a mean of three cigarettes per day. In the protocol, no cigarettes were mentioned. Two participants consumed small amounts of alcohol during the intervention two to six hours after training. The medium score in PASE was 138 (T_1_) and 134 (T_3_) which is categorized as “much physical activity”.

### 3.4. Responsiveness

Responsiveness was investigated qualitatively regarding nutritional stage, physical capacity, physical activity, disease stage, age, and mean oxygen saturation at activity. No special characteristics were found. There was a tendency of higher responsiveness with lower training intensity in the knee extensor. Women responded more than men in the knee extensor.

Considering only two participants (ID 1 and ID 3) were fasting under both conditions before training, good responsivity was seen. Results in the subjects with systemic or inhalative corticosteroids, underweight, and thyroid changes were not striking. With these results, different responses of the subgroups in terms of training intensity and nutritional state cannot be verified.

## 4. Discussion

The main outcome of the present study was the strength response to a meal rich in protein and carbohydrates. Comparing maximum strength after 24 h, results promoted the use of protein and carbohydrates after training. Differences in changes for maximum strength were significant for both leg extensor and chest press.

Both investigated measures showed significant interaction effects of food supplementation and post training performance. Food supplementation resulted in improved performance. At the same time, training alone did not result in any detectable exhaustion. The sizes of the interaction effects are conventionally large (ca. 0.9, chest) to very large (ca. 1.1, knee). The significance levels are comparably poor, however. This apparent contradiction can easily be explained by the small sample sizes and by the correspondingly large standard errors.

In fact, the diagnostic plots showed some lack of fit of the models. In particular, the Gaussianity of the residuals was not sufficiently met, while heteroscedasticity was fundamentally given. Usually, this condition results in conservative estimates of *p* so that the validity of the data analysis is unlikely to be impeded. Rather, it will be underestimated.

The Q-Q plots indicated bi-modality of the residuals rather than perfect Gaussianity. This might suggest influence of additional factors. These can be hypothesized to be given by sex, by degree of COPD or by many other physiological entities, but it must be noted that the residual standard deviations appear very small for both measures. The value of ca. ±1 kg in case of quadriceps performance looks reasonable to the sport scientist, but the equivalent of ca. ±0.4 kg for chest press might, however, indicate some over-fitting already. Consequently, we assume that the relevant factors have been sufficiently covered and that other factors presumably have minor effects.

Any attempt to incorporate more factors into the present data analysis must be doomed to failure anyway. The data set is simply too small. In fact, consideration of the factor sex, i.e., of merely one additional parameter estimate was technically impossible.

This was the first investigation measuring the short-term effects of nutritional intervention for patients with COPD who are habituated to training. Until now, no consistent significant change in strength was shown as add-on effects of nutrition in COPD [19]. Reasons might be in testing non-trained individuals since training itself is highly effective and might cover influence of nutritional support [21]. Add-on interventions might become more relevant in optimized training. Most evidence to reduce muscle dysfunction are suggested for nutritional interventions with high EEA [22], which is in line with study results. There is already evidence that nutritional intervention with protein and carbohydrates immediately after strength training might influence regeneration of strength in healthy people [16]. Training response in COPD is mostly preserved, while it has been shown that amino acid metabolism [25] and protein synthesis [24] are diminished in COPD compared to healthy adults. Training performance, however, vary to healthy adults since training performance is diminished which also results in smaller anabolic and catabolic response [24]. In previous studies, it was shown that additional intake of protein can increase protein synthesis in COPD [36,37]. In contrast to this, Constantin et al. did not find increased functional or molecular response to increased protein and carbohydrates after strength training [24]. For the recent study, a healthy control group doing same investigation would be necessary before specifying a statement about changes in strength response.

Different than assumed, maximum strength as well as physical performance in STST was mostly preserved or even increased after 24 h. These results are surprising since strength loss is the physiological answer to fatiguing exercises in healthy people. A supercompensation is not yet assumed after this short time. Different from previous studies [16,38], subjects were familiarized with training intervention, which can result in less strength loss. A possible explanation for increased strength values after training can be the short-term learning effect of the strength test since 1RM was not practiced in advance [39]. Nevertheless, for participating subjects, this should usually not occur due to their experience in strength training [40]. A clear explanation for strength gain was not found at this state of knowledge and must further be investigated.

There was a tendency to higher responsivity to lower training intensity, referring to 1RM in the knee extensor, which is in contrast to the previous literature [41]. In the study protocol, training intensity was determined with momentary muscle failure after one minute. This would suggest same intensity for all participants and might explain the findings. No other special characteristics were found for patients with similar conditions in nutritional stage, physical capacity, physical activity, disease stage, age, and mean oxygen saturation at activity. Women responded more to nutritional intervention than men, which is not in line with the previous literature [42]. Until now, there are no studies explicitly studying differences in training response between men and women in COPD [13]. Possible side-effects of confounding factors such as cachexia, high inflammatory state, or hormonal changes cannot be refused. To identify promoting or confounding variables, further investigations with larger samples are needed. Detailed information about health state, such as in this pilot study, are recommended to specify and analyze results as well as possible. It remains unclear which patients have a good responsivity to nutritional intervention.

Sour-milk cheese contained 36 g of protein and followed recommendations for older people that high amounts of 40 g protein increase response [26,43]. Since there is evidence that body size is not a determining factor for the amount of protein [26], the amount of provided per meal was not individualized; 53 g carbohydrates are lower than the recommendation of 1 g/kg body weight [14]. Proteins of animal sources with high BCAA content were used [26]. Carbohydrates were chosen after the recommendation of the simple, fast-absorbed type [44].

Analyzing available information about general nutrition, three participants most likely had reduced protein, carbohydrate, and total energy intake. Clear results are not possible because of little information in the food diary and its limited informative value.

In COPD, the overall prevalence of malnutrition is almost 20% [45]. In these patients, prevalence of sarcopenia is more than 70% and significantly higher than in people without malnutrition (12%) [45]. Nevertheless, sarcopenia can also occur with normal weight or obesity [46]. Sarcopenia and sarcopenic obesity are nutrition-related conditions [46], why eating habits should play an important role in treatment. Changed eating habits might be a reason for general malnutrition and lack of protein since only 65% in COPD can eat what they think is necessary [47].

The highest nutritional response was in two participants with probable imbalanced nutrition, while the lowest changes were in patients with very balanced nutrition. These findings must be handled with care because of the limited informative value of the food diary.

Three patients participated in training without any meal before, which is known to reduce performance [48]. In this pilot study, strength loss after training was not higher in fasting patients but responsivity was very good. Explorative analysis was very limited because of the small sample with additional confounding factors.

This pilot study was a first attempt to design a study with nutritional intervention focused on short-term strength response. Study feasibility was shown in advance and confirmed in this pilot study. Eligibility criteria were broadly defined, which reflects the heterogeneity of the disease. The results give a first reference to aspects that should further be investigated or probably play a secondary role. Inclusion of patients with psychic comorbidities other than depression should be discussed to avoid undesirable stress for those patients. Despite detailed information and dialogues, patients reported high exhaustion and worries about failing, which is why even more explanations or a practice run-through of the procedure can be considered for further studies. More repetitions with the same participants might reduce deviations and specify results. The effects of day condition and learning effects can be reduced that way.

Whole-food sources were used because of their content for other nutritional needs [14] and similar results to shakes [16]. This should suggest patients a proper diet in their everyday life that they can follow. Nevertheless, the high volume of the provided meal was reported to be not suitable for everyday by eight participants. Patients with COPD often have reduced appetite, insufficient energy for eating, and shortness of breath with eating [47]. Fluid or concentrated products might therefore be more suitable in following studies. Timing of the meal after training for best effects followed the best available knowledge for older people [49] but is still discussed [50]. In COPD, pre-exercise consumption of the same nutrients is suggested to be less tolerated by patients since large meals often cause dyspnea and this could affect training [47].

This study has several limitations. Subjects were not blinded which can affect results. Moreover, the study investigated a very small sample of great variety. Nevertheless, this variety was wanted since it represents patients who often have comorbidities and very different conditions. The good performance level in 6MWT and PASE is striking and might be different from the mean in patients with COPD. Results can be explained because all subjects are integrated in regular training. This was wanted since it is very likely that the training effect in the novice might cover the influence of nutritional support [21]. Good physical capacity promotes the integration in regular training in patients with COPD. Another limitation is the very little information about the general eating habits of participants. A standardized food protocol with information about protein and carbohydrate intake would be necessary for further analysis of general intake, malnutrition, and effects of usual nutrition. The study focused on the functional response to nutritional intervention. For more complex information about the underlying mechanism, the investigation of blood parameters such as serum insulin, blood glucose, serum cortisol, creatine kinase, and inflammatory markers such as interleukin 6, interleukin 10, and macrophage migration inhibitory factor are needed [16]. Further intervals for strength response can be analyzed in future to get more information about these processes. Moreover, long-term effects of training-specific nutritional interventions should be proved. Expanding investigations to muscle biopsy analysis with this intervention would widen understanding but must be discussed since this is suggested to reduce the willingness to participate.

The study also has strengths. There was a detailed reporting of training modality and sensitive strength test fitting to training type. All training variables were based on recent recommendations for strength training in COPD and the principle of local muscle fatigue, which is known to be the main indicator for effective training [51] and was proved for COPD [52]. In this study design, the functional outcome was measured, which is hypothesized to be the main outcome of patients. The 1RM is the gold standard [30,31], sensitive [53], and the main field test to assess peripheral muscle strength in COPD. It was assessed in the best investigated muscles of the upper and lower extremities in COPD [11,54]. The indirect method was used, which differs from other studies [31] but reduces the risk of high peaks in blood pressure and needs less practice and mental will [32]. It also allowed short testing procedures to avoid exhaustion by testing, which might have reduced performance in training. All outcome measurements refer to muscle dysfunction and consider muscle strength, muscle endurance, and exertion. Previous studies focused on structural and metabolic changes, which is often do not correlate with the patient’s functional capacity [55]. Nevertheless, additional information about laboratory parameters can be helpful for the understanding of underlying mechanisms.

## 5. Conclusions

Our results show that the ingestion of protein and carbohydrates immediately after strength exercise might affect maximum strength as a functional indicator for regeneration. These results are very promising and need further research. In the next step, general protein uptake in this target group should be assessed more detailed to get to know how general nutrition affects training response. The study needs a larger sample to confirm the results since this could result in higher training response and better training tolerance for patients.

## Figures and Tables

**Figure 1 nutrients-14-03565-f001:**
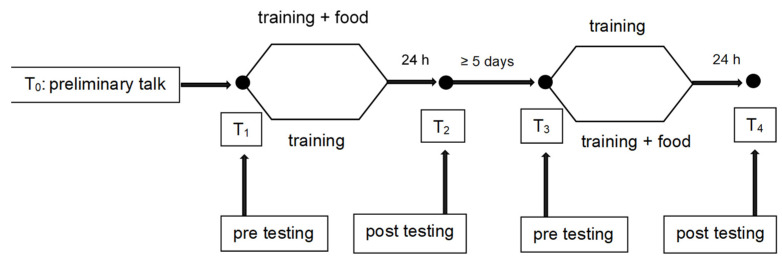
Cross-over design of intervention.

**Figure 2 nutrients-14-03565-f002:**
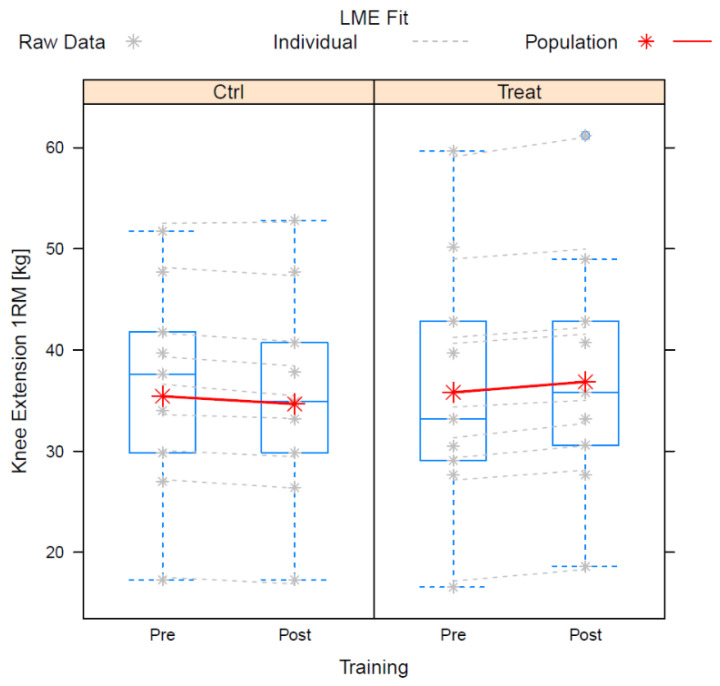
Boxplots of the raw data of the individual fits and of the population fits for knee extensor.

**Figure 3 nutrients-14-03565-f003:**
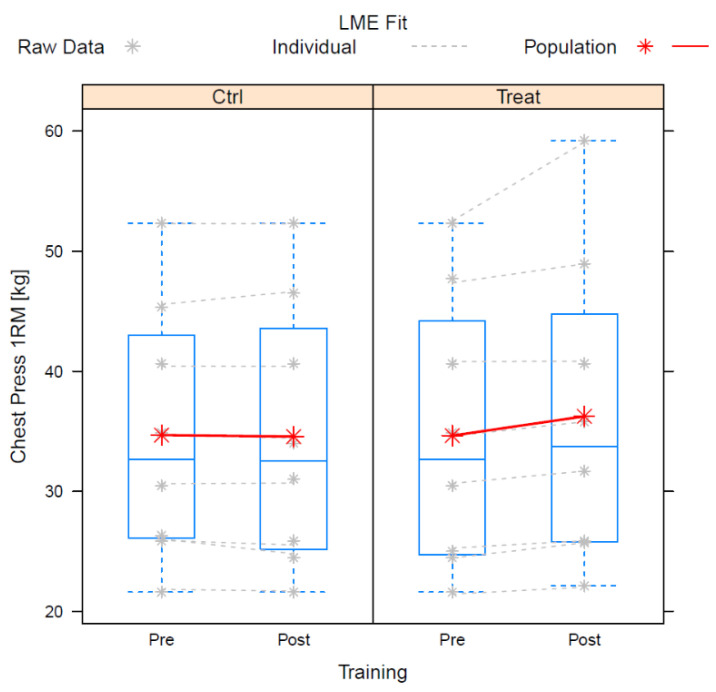
Boxplots of the raw data of the individual fits and of the population fits for chest press.

**Table 1 nutrients-14-03565-t001:** Definition of training variables.

Training Variable	Realization in Training Intervention
Muscle actions	Dynamic with 30% increased eccentric weight
Intensity (loading)	momentary muscle failure after 1 min
Volume	2 sets of machine circle with 8 machines 1 min of each exercise
Velocity of muscle action	Breathing frequency
Exercise selection (in order)	Electronic resistance machines (circle training): rowing, leg press, back extensor, abdominal flexor, knee flexor, knee extensor, bench press, lat pull-down
Rest period between sets	1–5 Min
Rest periods between exercises	1 Min

**Table 2 nutrients-14-03565-t002:** Nutritive value of provided meal.

Nutrient	Sour-Milk Cheese (100 g)	White Bun (60 g)	Total
Carbohydrates	<0.1 g	31.8 g	31.8 g
Protein	30 g	4.74 g	34.74 g
Fat	0.5 g	1.26 g	1.76 g

**Table 3 nutrients-14-03565-t003:** Coefficients of the linear mixed effects model (LME) fitted to the knee extension data. Estimates (Est.) correspond to one-repetition maxima [kg] and respective changes.

	Est.	Std. Err.	DF	*t*-Value	*p*-Value	Cohen’s d
(Intercept)	35.43	3.755	24	9.434	0.000	-
Training Post	−0.765	0.484	24	−1.58	0.127	−0.645
(Food)	0.384	0.983	24	0.391	0.699	0.16
Training Post × Food	1.804	0.681	24	2.65	0.014	1.082

**Table 4 nutrients-14-03565-t004:** Coefficients of the linear mixed effects model (LME) fitted to the chest press data. Estimates (Est.) correspond to one-repetition maxima [kg] and respective changes.

	Est.	Std. Err.	DF	*t*-Value	*p*-Value	Cohen’s d
(Intercept)	34.69	3.773	21	9.194	0.000	-
Training Post	−0.125	0.319	21	−0.391	0.699	−0.171
(Food)	−0.044	0.417	21	−0.105	0.917	−0.046
Training Post × Food	1.731	0.82	21	2.112	0.047	0.922

## Data Availability

The data presented in this study are available on request from the corresponding author.

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
