# Peer review of "Effects of Carbohydrate and Protein Administration by Food Items on Strength Response after Training in Stable COPD"

_nutrients, 2022, doi:10.3390/nu14173565_

Round 1
Reviewer 1 Report
Huhn and Diel performed a pilot study seeking to elucidate the effects of carbohydrate and protein administration on strength metrics following a bout of exercise training in participants with COPD. The current body of literature shows beneficial effects of carbohydrate and protein consumption following exercise in young and aged healthy participants. The pilot study nature of this investigation with participants diagnosed with stable COPD provides initial findings by which future research can be expanded upon.
Minor comments: There are small grammatical errors throughout that can be addressed. These errors are mostly misplaced commas and sentences that can be restructured to read clearer. Additionally tables for participant baseline characteristics and dietary recall information would be helpful (possibly supplementary data).
ABSTRACT:
Lines 13-15: numerous studies have shown ingestion of a meal rich in carbohydrate and protein following exercise improves recoverability and subsequent performance. These are not recent findings (studies ranging from the 1960s-present have shown this).
It would be beneficial to give a summary of the intervention treatments and training intervention in the abstract.
Introduction:
Lines 29-30: Could a percentage of deaths contributed to COPD be included in this statement. These data should be present on the world health organization website.
Lines 37-38: Please check the veracity of malnutrition and inactivity as the only definitive causes of muscle dysfunction. Numerous studies have shown other factors contribute to this, such as: chronic overtraining, muscle toxicity, inflammatory responses, etc.
Lines 67-69: The authors cite a study on high-intensity exercise; it would be tremendously helpful for this and other studies to cite whether the “high-intensity” exercise intervention was endurance (aerobic) in nature or resistance training. This will provide clarity for the reader.
Lines 87-90: Please insert the modality of training intervention used (i.e., high-intensity resistance training or high-intensity aerobic training). We know from the descriptor in the methods later in the manuscript this is a knee extensor exercise intervention but in eluding to this in the introduction helps to prime the reader.
Materials and Methods
Lines 163-168: The authors note that non-parametric statistics were used for analysis. It is unclear as to why this is the case. Were these data non-normally distributed or was this an a-priori decision?
For statistical analysis, was a 2x2 non-parametric model ran or was analysis done on change scores for each variable? It may be beneficial to run an ANCOVA type model (Quade’s test for non-parametric equivalent) using baseline or mean-centered baseline values as the covariate. Additionally, I would also entertain adding COPD grade and gender into the model as covariates.
Results:
Lines 205-207: While 0.72 is certainly a decent correlation between variables, generally a strong correlation is indicated by 0.8-1.0.
In the results, can the authors please provide 95% confidence intervals and effect sizes. This is particularly useful for future research so investigators can see if the trends they observe track with the data presented here. Additionally, can r^2 values be provided in addition to the r values for correlational analyses?
In the results section, I think the figures can be made more informatively. As presented, these appear as analysis done on change scores since there is no baseline value presented. The addition of the baseline values would also help interpretation. It is also useful and standard practice to include in your figure legend what the * values indicate.
Discussion
257-260: I’m not sure a comparison of response to healthy athletes can be made given that these two populations have never been tested side by side in a trial and that physiologically, these populations are distinctively different. I say this because the type of training intervention that would generally be used with an athletic population (or previously trained) is notably different than what was used here. Additionally, the nutritional requirements, body composition, etc., are very different. In fact, this is even mentioned in the successive sentences.
I found the discussion to be lengthy and scattered in thought (i.e., jumping back and forth between discussing strength adaptation and nutritional considerations). I think these parts can be condensed and focused in the clinical populous used as opposed to discussing other diseases such as diabetes.
In parts of the discussion statements of post exercise nutrition being absolutely beneficial for recovery of strength are made but with this sample size, those statements are very bold.
Author Response
Thank you for you annotation and recommendations to elevate the quality of our paper. We revised the paper, especially the statistics were overhauled with methods of higher value. Your annotation about the individual chapters were very helpful and we tried to get more clarity and structure with it.

Reviewer 2 Report
I firstly want to congratulate the authors on paper which is an interesting read.
My comments are in no way a criticism I merely want to elevate the quality of the paper.
I would suggest you proofread the document as some sentences lack clarity. I have outlined my comments below:
Line 19-24: Reword for clarity.
Line 33: Briefly explain the stages of COPD.
LIne 50: Damage is not reduced through ingestion of CHO and Protein, the recovery is quicker.
Line 57: chnage widespread to conflicted.
Line 60: reductions/increases in fat mass? What impications have this change?
Line 67: Chnage the start of the sentence to "Our research team have previosly/recently demonstrated that..."
Line 70: Explain 'shakes' or chnage to drinks/supplementation.
Line 112: Error message in the sentence.
Line 124: The sentence starts with a number.
Line 129: strength training program.
Line 130: Reword for clarity.
Lines 139-162: perhaps use sub-headings to seperate the protocols e.g. Strength testing/self-report measures/etc.
Lines 183 and 194: Medium change? Should this be median? If so explain why the mediam was used as opposed to the mean...
LInes 205-214: reword for clarity.
Lines 207 & 212: Error message in the sentence.
Line 215: You use different words to describe the food diary e.g. journal/diary of food chnage to food diary.
Line 216: More clarity about the food diary is needed in the methodology. Did they complete a food dairy or did they just record if they had 3 x meals per day including "1 warm meal?"
Line 259-263: Your point is very unclear. I wouldn't compare COPD patents to athletes. The training protocol used does not mimic any typre of athlete-appropriate strength trainig.
Line 289: You have already said and reiterate the low level of quality in the nutritional information collected so do not speculate that n=3 participants "were striking because of possible imbalanced nutrition..."
Line 294: Sarcopaenia is associated with inactitvity and aging. Are you trying to suggest that adequate nutrition could reduce sarcopaenia? if so state that.
Line 298: Again you have limited data on the nutrition habits of your cohort. Don't overstate your findings.
LIne 318-328: Suggest that future studies include a validated food diary/food frequency questionnaire. THis can be addressed in the limitations section.
Line 338: Could they consume a 'drink/liquid' during and after the training? Worth investigating in future studies...
LInes 379: Delete - "Follow up study is planned.
Author Response
Thank you for your nice and detailed feedback to the study. We could use many of your annotation with the aim to improve the paper. We did greater changes in the methods and overhauled the statistics with methods of higher value. We hope that after our revision you see more clarity in it, especially in the introduction and conclusion.

Round 2
Reviewer 1 Report
I’ve found the revisions presented herein by the authors to greatly improve the clarity and quality of the manuscript. All critiques brought forth have been addressed.
Reviewer 2 Report
Thank you for the responses see my comments below:
Line 33: Briefly explain the stages of COPD.
Explain the stages not all readers will be familiar with them. Explain stages 1-5...
LIne 50: Damage is not reduced recovery from the damage/stress/training is expedited... Explain
Lines 139-162: perhaps use sub-headings to seperate the protocols e.g. Strength testing/self-report measures/etc.
The section does not read 'fluently' as is. I would suggest sub-headings to guide the reader.
Lines 178-200: Write this section in paragraph form. It might have been the pdf file but the section was a series of one-sentence paragraphs.
Line 259-263: Your point is very unclear. I wouldn't compare COPD patents to athletes. The training protocol used does not mimic any type of athlete-appropriate strength training.
Thank you for this annotation. We agree with you that rationale was not well written. We added
evidence to make explanation clearer. Nevertheless, strength training could also be appropriate to
athletes since exercises were done until momentary muscle failure which is the best indicator for
effective training modality.
I would make a clear distinction between the training volume/quality/intensity of a COPD patient vs. an athlete, and the resultant muscle damage/fatigue and recovery required.
Please proofread the manuscript and fix the minor suggestions. Congratulations on an interesting and novel study.